# The Osteoarthritis Thumb Therapy (OTTER) II Trial: a study protocol for a three-arm multi-centre randomised placebo controlled trial of the clinical effectiveness and efficacy and cost-effectiveness of splints for symptomatic thumb base osteoarthritis

Jo Adams ![ORCID],[1] Paula Barratt,[1] Nigel K Arden,[2,3] Sofia Barbosa Bouças,[4] Sarah Bradley,[5] Michael Doherty,[6] Susan Dutton,[7] Krysia Dziedzic,[8] Rachael Gooberman-Hill,[9] Kelly Hislop Lennie,[1] Corinne Hutt Greenyer,[1] Victoria Jansen,[10] Ramon Luengo-Fernandez,[11] Claire Meagher,[1] Peter White,[1] Mark Williams ![ORCID] [12]

For numbered affiliations see end of article.

**Correspondence to**
Professor Jo Adams;
ja@soton.ac.uk

## ABSTRACT

**Introduction** The economic cost of osteoarthritis (OA) is high. At least 4.4 million people have hand OA in the UK. Symptomatic thumb base OA affects 20% of people over 55 years, causing more pain, work and functional disability than OA elsewhere in the hand. Most evidence-based guidelines recommend splinting for hand OA. Splints that support or immobilise the thumb base are routinely used despite there being limited evidence on their effectiveness. The potential effects of placebo interventions in OA are acknowledged, but few studies investigate the clinical efficacy of rehabilitation interventions nor the impact of any placebo effects associated with splints.

**Methods and analysis** Participants aged 30 years and over with symptomatic thumb base OA will be recruited into the trial from secondary care occupational therapy and physiotherapy centres. Following informed consent, participants will complete a baseline questionnaire and then be randomised into one of three treatment arms: a self-management programme, a self-management programme plus a verum thumb splint or a self-management programme plus a placebo thumb splint. The primary outcome is the Australian Canadian Osteoarthritis Hand Index (AUSCAN) hand pain scale. The study endpoint is 8 weeks after baseline. Baseline assessments will be carried out prior to randomisation and outcomes collected at 4, 8 and 12 weeks. Cost-effectiveness analysis will be conducted and individual qualitative interviews conducted with up to 40 participants after 8 weeks to explore perceptions and outcome expectations of verum and placebo splints and exercise.

**Ethics and dissemination** South Central—Oxford C Research Ethics Committee approved this study (16/SC/0188). The findings will be disseminated to health professional conferences, journals and lay publications

## Strengths and limitations

► This trial is powered to evaluate the clinical benefit and statistical significance of adding splinting to a self-management programme for people with thumb base osteoarthritis (OA).
► This is the first trial to use a placebo thumb splint intervention and will add to the understanding of contextual aspects of such visible physical treatment and self-management programmes for OA.
► The trial has been informed and designed with the input of patients and expert clinicians and the trial outcome measures have been agreed as meaningful by patients.
► The 12-week trial follow-up period is limited and longer term follow-up would provide further useful outcome data.

for patient organisations. The research will contribute to improving the management of thumb base OA and help clinicians and patients make informed decisions about the value of different interventions.

**Trial registration number** ISRCTN 54744256.

## INTRODUCTION

Osteoarthritis (OA) is a leading cause of pain, disability, healthcare utilisation and productivity loss in the UK. Each year approximately two million adults visit their general practitioner (GP) with symptoms of OA.[1] OA is more prevalent in women, the incidence increases with age and recent estimates suggest more than 150 million

Europeans have radiographic hand OA and 15 million have symptomatic arthritis.[2] Symptomatic thumb base (first carpometacarpal and/or scaphotrapezial joint) OA affects approximately 22% of people aged 50 years and over.[3] Thumb base OA is likely to become more prevalent in the future since the incidence increases with age[4] and is identified as a priority for treatment as it causes more pain and disability and is associated with a worse prognosis than other hand sites affected by OA.[5]

Patients with thumb base OA present with predominantly mechanical usage-related pain over the thumb base[6] and are more likely to have more pain, work disability and reduction in quality of life and function, and to receive more anti-inflammatory drugs and more splinting than participants with OA affecting other hand joints[5 7–9]

Thumb base OA affects entire hand function[10] and the overall impact, particularly in older people, can be substantial, with many experiencing difficulties with daily household, caring, work and leisure activities.[11] Despite the scale of this problem, it appears that both patients and practitioners often believe that there is little that can be done.[12]

Therapeutic splinting for thumb base OA aims to minimise or eliminate motion at the thumb carpometacarpal joint (CMCJ)[13] in order to prevent joint deterioration and/or deformity, decrease pain and increase overall hand function.[14] The process of designing thumb splints currently lacks detailed reporting but biomechanical principles have been applied to one design to off-load the dynamic forces occurring during functional hand use on a symptomatic CMCJ.[15] It is known that stabilisation of the CMCJ to reduce pain levels impacts on hand functionality.[16] Currently, thumb splints are recommended for patients with thumb base OA.[17] However, the efficacy of splinting based on this approach has not yet been established and forms the rationale for this trial. Some evidence based on small, non-powered samples[18 19] show thumb splints can help relieve pain; however, evidence to support their effectiveness is not yet fully supported by robust research.[20] There is insufficient evidence to suggest that the combination of splinting delivered alongside hand exercises is more effective than hand exercises alone[21–23] and the evidence to support splinting in alleviating hand pain in the medium term is supported by low-level evidence only.[24] To date, there has also been no examination of any contextual and non-specific patient–practitioner interaction effects associated with splinting in thumb base OA. If this is substantial, optimising the non-specific effects of treatment could improve treatment effectiveness and overall care of people with OA in a safe and cost-effective way.[25 26] This randomised controlled trial (RCT) protocol was designed in accordance with the Standard Protocol Items: Recommendations for Interventional Trials checklist,[27] to evaluate the effectiveness, efficacy and cost-effectiveness of thumb splints for people with thumb base OA.

The primary objective of this trial is to determine the clinical effectiveness and efficacy of thumb splints when added to a self-management programme for people with symptomatic thumb base OA. The trial questions:

1. Is there a benefit of adding a thumb base splint to a self-management programme for people with thumb base OA?
2. Is there a difference in benefit between adding a verum or a placebo thumb base splint to a self-management programme for people with thumb base OA?
3. What are patients' views and experiences of the effectiveness, acceptability and adherence to the trial interventions?

## METHODS AND ANALYSIS
### Patient and public involvement
Exploring the effectiveness of orthotic devices for use by patients living with OA was identified as a priority area for investigation by an Arthritis Research UK patient stakeholder committee. Two patient and public group meetings were carried out with eight patient partners living with hand OA to listen to patient experiences of living with hand OA, explore intervention components of the trial and help to inform co-design a placebo splint design.[28] Patient partners discussed with the team what splints they thought should be included in a trial, which splint designs they found most credible and which outcomes were important to them. These discussions informed the study design. A national Delphi study with therapy clinicians and patients was conducted, to inform and agree trial processes.[29 30] Clinicians tended to want more focused time efficient measures where patients usually hoped for more comprehensive intervention that was beyond the time allocated to National Health Service (NHS) outpatient provision. This helped to consider treatment burden from both clinical and patient's view points. This involvement at an early stage ensured that clinicians felt that they had contributed and had ownership in the trial design and patient input ensured that patients' perspectives had been integral to the study design hopefully making the trial relevant and appealing to prospective patients. We conducted a focus group to explore the support required by NHS therapy clinicians when knowingly delivering placebo splints.[31] A pilot study across five NHS recruitment sites to test recruitment and procedural feasibility and safety and the convincing delivery of a newly designed placebo splint was conducted and reported.[32] The education and support needs identified by therapy clinicians taking part in a placebo controlled RCT was sought and recorded.[33] This informed the trial training for clinicians. Our patient partner and co-author (CHG) provided a patient's perspective in developing recruitment strategies, study conduct and lay dissemination routes.

### Trial design and setting
The Osteoarthritis Thumb Therapy (OTTER) II Trial is a pragmatic, multi-centred, single (participant) blind,

superiority randomised controlled clinical effectiveness and efficacy trial. People with symptomatic thumb base OA reporting moderate to severe thumb base pain, will be equally allocated to one of three groups: Group A: 8 weeks of a facilitated self-management programme, Group B: 8 weeks of a facilitated self-management programme plus a verum splint or Group C: 8 weeks of a facilitated self-management programme plus a placebo splint. All groups will be encouraged to continue their self-management and, where appropriate, splint wear until follow-up at 12 weeks from baseline. The study will run from 28th February 2017 to 14th March 2019.

Participants will be recruited consecutively from referrals to occupational therapy or physiotherapy (Therapy) departments from new, current and review patients of Rheumatology, Orthopaedic, Hand Surgery Units and General Practice at 17 NHS recruitment sites (online supplementary appendix 1). Intervention will be delivered by a qualified OTTER II Trial trained occupational therapist or physiotherapist who has worked independently in a clinical role treating patients with hand OA and who works within a therapy department that accepts referrals for patients with thumb base OA.

### Participants

Participant inclusion criteria were decided on iteratively with our rheumatologists, clinicians and hand surgeons in relation to the literature about prevalence, incidence rates and current treatment by therapy departments in the UK. Clinical tests, that contributed to inclusion criteria, were selected following: review of literature; consideration of the sensitivity and specificity of each test; examination of the practicality and feasibility of the testing procedure for the collaborating clinicians working across different UK hospitals. As there is no clear consensus on the longevity of the impact of intra-articular steroid injections into the first CMCJ,[34–36] we used local clinical NHS protocols relating to the length of time suggested to repeat first CMCJ steroid injections, to inform our inclusion criteria.

Consecutive potential participants will be screened and assessed for recruitment into the trial by the collaborating the OTTER II Trial trained therapy clinicians using inclusion and exclusion criteria as shown in table 1. The characteristics of participants who fulfil inclusion criteria and who decline to take part will be recorded.

### Interventions

The trial's three intervention arms all include 90 min of direct therapy intervention delivered by a qualified OTTER II Trial trained occupational therapist or physiotherapist who has worked independently in a clinical role treating patients with hand OA and who works within a therapy department that accepts referrals for patients with thumb base OA.

| Table 1 | OTTER II Trial patient inclusion and exclusion criteria |
|---|---|
| **Inclusion criteria** | |
| 1 | Aged 30 years and over |
| 2 | At least moderate hand pain (AUSCAN[42] hand pain score >5) and moderate functional hand disability (AUSCAN[42] hand functional disability score >9) |
| 3 | Show signs and symptoms of thumb base OA on clinical enquiry and examination, specifically: hard tissue enlargement of the first CMCJ OR squaring of the thumb base OR pain that worsens when pinching OR pain that worsens on span grip (eg, opening a jar) OR crepitus on movement OR reduction in thumb base range of movement OR positive thumb adduction provocation test[52] OR positive thumb extension provocation test[52] OR pain on palpation of the dorso-radial aspect of the thumb CMCJ |
| 4 | No other household member participating in the trial |
| 5 | Able to give written informed consent |
| 6 | Available to attend occupational therapy/physiotherapy/hand therapy sessions |
| **Exclusion criteria** | |
| 1 | Consultation with therapy department or treatment for this thumb problem (excluding pain killers and anti-inflammatories) in the previous 6 months |
| 2 | Intra-articular joint injection to wrist, fingers or thumb in the previous 2 months |
| 3 | Fractures or significant injury or surgery to the wrist or hand within the previous 6 months |
| 4 | Red flags. History of serious illness or disease such as any other diagnosed rheumatic conditions: gout, psoriatic arthritis, ankylosing spondylitis, connective tissue disorders (systemic lupus, systemic sclerosis), resulting in inflammatory arthritis in the hand/s, or, progressive neurological signs, or acute swollen hand joint |
| 5 | Diagnosis of dementia or other significant disorder likely to affect communication |
| 6 | Already received thumb splints for thumb base OA |
| 7 | Skin disease that may interfere or contraindicate splint wear |
| 8 | Participant of a drug or medical device trial in the last 12 weeks |

CMCJ, carpometacarpal joint; OA, osteoarthritis; OTTER Trial, Osteoarthritis of the Thumb Therapy Trial.

The direct therapy intervention will be delivered in a 60 min baseline appointment at an NHS secondary care hospital or clinic, when the interventions listed below will be delivered. A 30 min follow-up intervention at week 4 will be conducted, where progress is reviewed and any necessary adjustments made. A final third hospital visit at week 8 is for finalisation of trial procedures only and includes no direct therapy intervention.

The three intervention arms, delivered at the baseline appointment are:

### Group A: optimal self-management programme
The self-management programme (online supplementary appendix 2) includes:
1. Teaching standardised hand exercises (developed from an evidence-based review[37]) and provision of a trial-specific hand exercise booklet.
2. Provision of a trial-specific booklet about joint protection, activity pacing and general advice about OA followed by a discussion with the therapist of the content.
3. Provision of the Arthritis Research UK Osteoarthritis information booklet.
4. A discussion with the therapist of the facilitators and barriers to engaging with self-management principles.
5. A patient hand exercise diary.

### Group B: optimal self-management programme plus a verum thumb base splint
Group B participants receive:
1. The optimal self-management programme as detailed above in Group A.
2. A verum splint. Therapists and participants will be given an option of two different splints, with the choice guided by a standardised Splint Decision Protocol (online supplementary appendix 3).

The splinting options informed by the study's design and development stage will be:
▶ Procool Thumb CMC Restriction splint (black) (Ref PTRS).
  Or
▶ Orfilight 2.5 mm 3/32" microperforated (beige) trouser leg splint (custom made by the therapist from a pre-cut standardised trial template and standardised strapping protocol)..

Both splints will be delivered with instructions on how to wear and use the splint (online supplementary appendix 4). The Procool thumb CMC restriction splint comes in packaging with a label providing details of the manufacturer and washing instructions.

3. A discussion with the therapist of the facilitators and barriers to engaging with splint wear (online supplementary appendix 5).

4. A patient splint wear diary (online supplementary appendix 6).

### Group C: optimal self-management programme plus a placebo thumb base splint
Group C participants receive:

1. The self-management programme as detailed above in Group A.
2. A placebo splint. There will be the choice of two designs of placebo splint. These are made in a lightweight nylon, secured around the wrist and have phalangeal components but no basal thumb joint support. One is black and the other is beige to match the verum splint options and three sizes will be manufactured. They have been designed with no known active component and none was detectable during testing.[38] A standardised Splint Decision Protocol about which splint will be most appropriate to issue in which situation will be used (online supplementary appendix 7). The placebo splints arrive in packaging with a label providing details of the manufacturer, washing instructions and a lifestyle education leaflet about how to position and wear the splint. Both splints will be delivered with instructions on how to wear and use the splint.
3. An information sheet to outline when the splint should be worn (online supplementary appendix 4)
4. A discussion with the therapist of the facilitators and barriers to engaging with splint wear (online supplementary appendix 5).
5. A patient splint wear diary (online supplementary appendix 6)

Trained OTTER II Trial clinicians will deliver a standardised self-management package of care using their clinical judgement to apply the content for each individual patient. Participants in group A will receive more time spent on self-management than participants in groups B and C.

### Concomitant care
Any relevant contralateral thumb treatment for participants will be delayed until the end of the trial. There will be no alteration in participants' general concomitant care while on the trial. Additional treatments, for example, joint injection, surgery and reported purchase or use of own splints during the study period will be captured on self-report questionnaires. Criteria for modifying allocated interventions are detailed within the OTTER II Trial Safety Information.

### Investigator training
Trial training visits will be conducted by the OTTER II Trial management team (JA, PB and PW) to carry out standardised instruction and demonstration for OTTER II Trial research clinicians prior to the start of patient recruitment. Training will cover procedures for maintaining clinical assessor and participant blinding, delivery of eligibility tests, interventions, reporting of safety events, data entry and use of case report forms (CRFs). Sites will also be visited by the trial team (JA, MW and CM) to provide support and guidance on maximising participant recruitment through liaison with surgery and community health teams, and to conduct quality assurance evaluations of intervention delivery as required.

A page of the trial website which is only available to site staff (not participants or the public) will provide training videos on the standardised delivery of all the trial interventions. All clinicians will complete the National Institute of Health Research Good Clinical Practice (GCP) training in order to have the skills and knowledge necessary to comply with the international ethical, scientific and practical standards to which all clinical research is conducted. Compliance with GCP provides public assurance that the rights, safety and well-being of research participants are protected and that research data are reliable.[39]

### Participant identification and baseline assessment

Potential participants will receive a letter about the study and a participant information sheet (PIS) at the hospital clinic or by post. For interested patients an outpatient therapy department appointment will be made at week 0 (baseline). Following eligibility screening, formal written consent will be obtained and participants will complete the baseline assessments. Consent will be obtained by an NHS therapist trained in OTTER II Trial procedures.

### Randomisation

Eligible patients will be enrolled into the study via the Oxford Clinical Trials Research Unit (OCTRU) online randomisation service and this system will record eligibility and stratification data.

When participants are randomised to a treatment arm, this constitutes the date of start of treatment. Participants will only be randomised once and for the treatment of one thumb. For participants with symptoms of bilateral thumb base OA, the most painful thumb will be treated first within the trial. If at the end of the trial participants require treatment on a contralateral thumb, then participants will be invited back by the clinical service to undergo routine (ie, non-trial) clinical intervention.

Randomisation will be on a 1:1:1 basis. The first 30 participants will be randomised using a simple random list with varying block sizes to seed the subsequent minimisation algorithm. This random list is generated by the trial statistician and concealed from all other members of the trial team to ensure future treatment allocations cannot be predicted. Subsequent participants will be randomised using a validated computer randomisation programme (Registration / Randomisation and Management of Product (RRAMP)) with a minimisation algorithm, which will ensure balanced treatment allocations across the stratification factors (centre, baseline AUSCAN hand pain score (Category 1=AUSCAN pain scores 6–12 and Category 2=AUSCAN hand pain scores 13–20) and treated hand dominance). The minimisation system includes a random element (0.8) to minimise predictability of treatment allocations for new participants.

Participants will be blind to treatment allocation. Strategies to maximise participant blinding include training for trial clinicians in how to present and discuss the treatment arms positively without disclosing other treatment options and routine assessment included in the quality assurance trial monitoring visits of clinician's communication and interaction in presenting treatment options with participants. The PIS has been carefully worded to state that an element of the self-management intervention may be placebo but does not state what aspect this may be. The study will be described as a comparison of self-management interventions and not divulging detail about placebo splinting in any clinician communication. Trial participants will not receive group intervention and therefore cannot compare their interventions received and we will not include participants who have already received thumb base splints. All of these strategies will help to maintain participant blinding. Contamination between treatment arms will be limited by identifying on hospital notes that the patient is part of the OTTER II Trial and therefore should not receive any further hand therapy treatment from the hospital team. It is not possible to prevent participants allocated to arm A or C purchasing their own splints during the trial period; however, the data will be captured in the final study questionnaire and will be used to assess any degree of contamination. The OTTER II Trial placebo splints will not be able to purchased privately.

This approach will support participant blinding, in addition to the training that treating therapists will receive about delivering placebos and responding to participant questions. The success of participant blinding will be assessed via patient questionnaires at 12 weeks.

### Treatment and follow-up

Figure 1 flowchart shows the participants' progress through the trial. After randomisation, participants will receive their allocated intervention during a 60 min appointment (week 0). All participants in each treatment arm will receive a 10 min telephone call at 2 weeks from the therapist to check for adherence to the self-management programme and to discuss any problems identified by the participant as relevant. In addition for Groups B and C participants, the therapist will check splint wear and comfort. Participants will attend for a 30 min appointment 4 weeks after randomisation, to review treatments and make adjustments where necessary and an assessor blind to treatment allocation will administer the objective Grip Ability Test[40]. A final hospital visit occurs 8 weeks after randomisation. A paper questionnaire will be posted to each participant at 12 weeks to collect follow-up patient-reported outcomes. After the 12-week follow-up has been completed (study endpoint), all patients will be offered the option of a further follow-up appointment for any further care that is needed. If patients have received verum splints, then they may keep these, if patients have received placebo splints or self-management alone then

OTTER II Trial: flow of participants

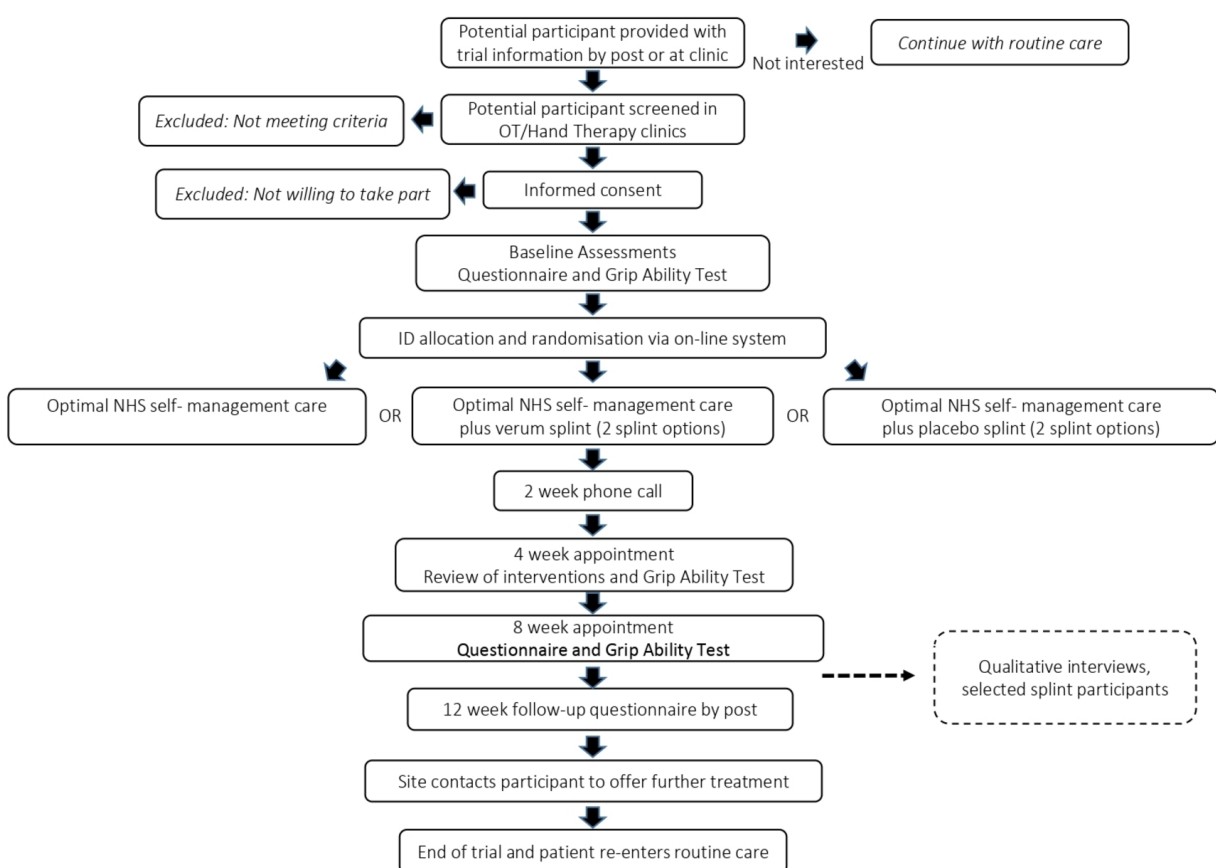

**Figure 1** Participants' process throughout the trial. NHS, National Health Service; OT, occupational therapy; OTTER II Trial, Osteoarthritis Thumb Therapy II Trial.

they will be offered the opportunity to receive a verum splint if required.

## Qualitative interviews

Between weeks 8 and 12, a qualitative telephone interview will take place for a subgroup of patients in intervention arms B and C. The researcher will be blinded for the qualitative interviews to prevent bias related to treatment allocation during data collection. In order to make this practical, the researcher carrying out data collection will be independent from the RCT and will be unfamiliar with the splints and differences between them. The methodological approach for the qualitative study will be a framework analysis. Purposive sampling will be used and based on male to female ratio (as per the study sample); age and AUSCAN hand pain index at baseline. Interviews will be conducted with up to 40 participants. Interviews will be semistructured, using a topic guide informed by patient partners, to elicit experience of splint and self-management interventions, splint preference, adherence and views about reasons for effectiveness. There will be a password for each participant that has agreed to take part in the interview study so that the participant can be correctly identified on the telephone. Consent for participation in

the qualitative interview will be sought and given at the time of entry into the main study.

The schedule of enrolment, interventions and a summary of assessments are shown in table 2. Detailed information about baseline assessments and outcome measures are listed in table 3.

The Global Assessment of Change[41] question adapted for the OTTER II study asks 'With respect to your thumb base pain how would you describe yourself now as compared with the start of your OTTER Trial therapy treatment'. The answer is given on a five-point Likert scale, that ranges from 'very much worse', 'worse', over 'no change' to 'better' and 'completely recovered'. The AUSCAN hand index for pain[42] recorded at baseline and 8 weeks is the primary outcome measure. The CRFs for this study can be obtained from the first author (JA)

## Sample size

The sample size has been calculated based on undertaking a global analysis of covariance (ANCOVA) for the primary outcome, AUSCAN hand pain[42] at 8 weeks, adjusting for the baseline pain score and stratification factors (including centre and treated hand dominance) across all three treatment arms using the power and sample size package, PASS 11 (PASS 11. NCSS, LLC.

**Table 2** Schedule of enrolment, interventions and assessments

| | Study period | | | | | | | Close-out |
|---|---|---|---|---|---|---|---|---|
| | **Enrolment** | **Allocation** | **Post-allocation** | | | | | |
| Time point: | **Baseline day 0** | **Baseline day 0** | 2 weeks | 4 weeks | 8 weeks | **9–11 weeks** | 12 weeks | **After 12 weeks** |
| Enrolment: | | | | | | | | |
| Eligibility screen* | X | | | | | | | |
| Informed consent* | X | | | | | | | |
| Randomisation/allocation | | X | | | | | | |
| Interventions: | | | | | | | | |
| Optimal NHS self-management care | | X | | | | | | |
| | ◆━━━━━━━━━━━━━━━━◆ | | | | | | | |
| Optimal NHS self-management care plus verum splint | | X | | | | | | |
| | ◆━━━━━━━━━━━━━━━━◆ | | | | | | | |
| Optimal NHS self-management care plus placebo splint | | X | | | | | | |
| | ◆━━━━━━━━━━━━━━━━◆ | | | | | | | |
| Telephone call check of progress | | | X | | | | | |
| Review of self-management care and splint wear (if applicable) | | | | X | | | | |
| Assessments: | | | | | | | | |
| Baseline assessments: see table 3 | X | | | | | | | |
| Outcomes: Grip Ability Test[40] | | | | X | X | | | |
| Outcomes: see table 3 | | | | | X | | | |
| Outcomes at follow-up: See table 3 | | | | | | | X | |
| Exercise adherence | | ◆━━━━━━━━━━━━━━━━━━━━━━━━━━◆ | | | | | | |
| Splint adherence | | ◆━━━━━━━━━━━━━━━━━━━━━━━━━━◆ | | | | | | |
| Qualitative interviews (selected participants) | | | | | | X | | |
| Patient request for further treatment | | | | | | | | X |

*The eligibility screen and informed consent can optionally be carried out prior to time point 0.
NHS, National Health Service.

Kaysville, Utah, USA. www.ncss.com). Assuming 80% power, a 5% two-sided significance in order to detect a standardised mean difference of 0.4 (a moderate effect size[43] based on a difference in the AUSCAN hand pain score of 2 points) and assuming a SD of 5, based on data from the OTTER pilot study,[31] requires 92 participants per arm. Allowing a 20% loss to follow-up at 8 weeks inflates this to 115 participants per arm, giving a total of 345 participants.

The sample size has taken into account the global comparison of the null hypothesis that there is no difference between the three treatment arms, and pairwise comparisons will only be undertaken if this global comparison is statistically significant. No further adjustment for the sample size has been undertaken to allow for multiple testing.

Clustering is not a consideration in sample size as 'Centre' is a stratification factor, and although there may be more than one therapist per centre, this should largely ensure balance across treatment arms within each centre. The Data and Safety Monitoring Committee (DSMC) (online supplementary appendix 8) will review the assumptions underlying this sample size calculation after approximately 50% of the participants have been recruited and followed up for 8 weeks, if still within the recruiting period.

### Recruitment strategy
Recruitment targets and procedures have been tested in the pilot study[32] and are based on these data. Recruitment start dates for the collaborating sites will be staggered over a 6-month period.

### Data collection, management and analysis
The trial will use a series of CRFs to record trial activities and RCT site staff will ensure that each CRF is completed properly and stored in the investigator's site file (ISF). Site staff will make photocopies and post CRFs to the

**Table 3** Baseline assessment and outcome measures

| Measure | Screening | Baseline | 4 weeks | 8 weeks | 12 weeks |
|---|---|---|---|---|---|
| Work Productivity and Activity Impairment Questionnaire)[48] | | ✓ | | ✓ | ✓ |
| Generic Quality of Life (SF12-V2)[53] | | ✓ | | ✓ | ✓ |
| EuroQol 5 Dimensions 5-Levels questionnaire[46] | | ✓ | | ✓ | ✓ |
| AUSCAN hand stiffness[42] | | ✓ | | ✓ | ✓ |
| Michigan Hand Questionnaire[54] | | ✓ | | ✓ | ✓ |
| Thumb pain over the last week | | ✓ | | ✓ | ✓ |
| Disability of the arm, shoulder, hand questionnaire[55] Leisure section only | | ✓ | | ✓ | ✓ |
| Arthritis Self-Efficacy Pain Scale[56] | | ✓ | | ✓ | ✓ |
| AUSCAN hand pain[42] | ✓ | | | ✓ | ✓ |
| AUSCAN hand function[42] | ✓ | | | ✓ | ✓ |
| Global assessment of change[41] | | | | ✓ | ✓ |
| Health utilisation questionnaire | | | | ✓ | ✓ |
| The objective clinician assessed Grip Ability Test[40] | | ✓ | ✓ | ✓ | |

OTTER Trial team using freepost envelopes, according to a standard operating procedure (SOP) so that both sites and the trial centre will have a copy of CRFs.

Participants will complete questionnaires at baseline, 8 weeks and 12 weeks. Participants will be asked to record the frequency for which hand exercises were completed for at least 20 min using a daily diary. Splint wear diaries will be used to capture hours per day of splint wear and will contribute to per protocol analysis of data. Participants will be provided with freepost envelopes in order to return questionnaires and diaries to the trial centre.

Where postal follow-up questionnaires are not received when expected, first they will be chased by post, and if there is no response key endpoint data will be collected over the phone. Up to three phone calls will be made to obtain key endpoint data. If participants wish to withdraw from their randomised intervention, they will still be included in the trial follow-up, unless they request to be excluded from follow-up. Where available, reasons for withdrawal from follow-up will be collected.

The OTTER Research Team will make copies of CRFs, questionnaires and diaries created by or received directly at the trial centre and post them to the relevant site to be filed in the ISF. The original documents will be secured in the Trial Master File.

Eligibility and stratification data will be entered directly onto the OCTRU online randomisation system by site staff. All other data will be entered into the trial database by the OTTER Research Team. The OTTER Trial team will check each CRF for completeness and where appropriate contact the site for missing data. A data management and sharing plan contains fully comprehensive information about all aspects of data management. All data will be processed according to the Data Protection Act 2018. All study-specific documents, except for the signed consent form and letters to participants, will refer to the participant with a unique study participant number/code and not by name.

Quantitative data will be stored on an OpenClinica trial-specific database prepared and managed by OCTRU. Qualitative data will be stored in QRS NVivo and word documents. The database has inbuilt data validation checks and a trial management system for managing data queries. Peer review of data entry will be conducted on 5% of CRFs. Data discrepancies will be reviewed on a regular basis to help clean the data. All data will be securely stored only accessible by authorised personnel agreed by the Principal Investigator, Legal services at the University of Southampton and The OTTER II Trial Steering Committee. Data will be backed up, and participant identifiable data will be stored separately from study data. Trial documentation will be retained for 10 years after completion of study-related activities and managed in accordance with the University of Southampton and the University of Oxford research data management policies.

Audio-recordings from the interviews will be transcribed verbatim, and participants allocated an ID number. The text of the interviewer notes from the telephone interviews will be anonymised and linked to the interview through the ID number. Names of participants, the names of any people discussed, healthcare employees and hospitals will be removed from transcripts and replaced with pseudonymns. Participants will be sent a letter to thank them for taking part in the telephone interview. A 'future use of interview' question is included on the consent form which gives copyright to the Universities of Southampton and Oxford to use the material in research, teaching, publications and broadcasting.

## Statistical analysis

The statistical analysis will be carried out by the OTTER II Trial OCTRU trial statistician. As is usual practice for the Oxford Cinical Trials Unit the statistician will not be blind to treatment allocation. There are rigourous checks and balances in place to ensure that the trial statistician cannot bias outcome.

This three-arm trial will first assess the global comparison of differences between the three treatment arms at the 5% level. The primary (global) null hypothesis is that there is no difference between the three arms. Only if this null hypothesis is rejected at the 5% (two-sided) significance level, will the pairwise comparisons be carried out in order to explore where the difference lies:

► Verum splint +self-management programme (Group B) versus self-management programme alone (Group A)
► Placebo splint +self-management programme (Group C) versus self-management programme alone (Group A)
► Verum splint +self-management programme (Group B) versus placebo splint +self-management programme (Group C)

It is anticipated that there will be an adjustment for multiple testing at this stage, and Bonferroni or a less conservative method will be used. The main analysis will be intention to treat. A per-protocol analysis of the primary endpoint will be performed as part of the sensitivity analyses. A statistical analysis plan will provide full details of all planned analyses.

The statistical analysis of the primary outcome, AUSCAN hand pain score, will be performed using ANCOVA, adjusting for baseline pain and stratification factors. Continuous secondary outcomes will be analysed using similar methods to the primary outcome. The unadjusted secondary binary variables will be compared using $\chi^2$ tests with logistic regression being used to adjust for stratification and important prognostic factors in a multi-variable framework.

Baseline characteristics for the three groups will be presented using the appropriate descriptive statistics. Baseline characteristics of participants who completed the trial and those who dropped out will be presented and explored in order to ascertain patterns of loss to follow-up. Subgroups will be examined to look for consistency of any observed treatment effects (using interactions). This analysis will be considered of an exploratory nature potentially providing hypotheses for future studies. The study is not powered to examine these in detail and will not therefore report p values for these.

## Qualitative analysis

In all, 40 interviews will be carried out by a researcher blind to participant treatment allocation.

Audio-recordings of the 40 interviews will be transcribed verbatim, anonymised and imported into analysis software QRS NVivo. The researcher will conduct primary coding of the narrative data and the five stages of framework analysis as described by Ritchie and Lewis[44] will be followed: familiarisation, identifying a thematic framework, indexing, charting and mapping/ interpretation.

Using a thematic approach,[45] data will be inductively coded, codes grouped to create categories and a descriptive account will be produced. Up to 10 interviews will be double coded to increase the validity of the findings. Double coding means more than one researcher independently assigning pre-specified codes to the qualitative data. NVivo software (QSR International) will be used to facilitate the double coding process. This allows for coding comparisons to be run between different coders. The results are returned according to the kappa coefficient score and the percentage of agreement between coders. After double coding transcripts, we plan to use this coding comparison for discussion and reflection on the data. The numeric measure will be used as a method of comparison to gauge if there is agreement and understanding of the definitions of the codes between team members. Any discrepancies identified will then be discussed until an agreement is reached and any differences resolved. The account will provide structured descriptions of participants' experience of self-management and splints, including reasons for their views about effectiveness as well as information about acceptability and adherence.

## Quality of life analysis

The EuroQol 5 Dimensions 5-Levels (EQ-5D-5L) questionnaire[46] will be administered to participants at baseline, 8 weeks and 12 weeks. Responses will be converted into utilities using tariffs estimated from a representative sample of the UK population.[47] Survival information collected from the trial will be combined with EQ-5D utilities to generate Quality Adjusted Life-Years (QALYs).

## Economic evaluation and analysis

The perspective adopted in the economic analysis will be that of the NHS. Costs associated with the following healthcare resource categories over the 8-week intervention period and follow-up period (12 weeks) will be included.

► Intervention provision (including splints and clinical staff time required for splinting); and
► Primary care contacts, including surgery and home visits by GPs, nurses, and out-of-hours medical services, and community therapists; and
► Hospital care services, including scheduled and unscheduled inpatient admissions, surgery, accident and emergency visits and outpatient care contacts.

Primary and hospital care resource use will be obtained from patient questionnaires administered at 8 weeks and at 12-week follow-up. Healthcare resources will be valued using unit cost schedules such as Personal Social Services Research Unit and NHS Reference costs. Costs associated with splints and other disposables will be obtained from the manufacturers and the NHS Supply Chain catalogue. Using the Work Productivity and Activity Impairment

Questionnaire,[48] the number of work days lost by study participants and the impact that OA had on the levels of productivity/activity and unpaid work will also be measured over both study periods.

An economic evaluation adherent to guidelines for good economic evaluation practice[49] will be undertaken. A within-trial cost-utility analysis will explore the incremental cost per QALY gained by splinting of the thumb base and a self-management programme when compared with (1) placebo-splinting and a self-management programme and (2) a self-management programme alone. The analyses will be conducted at 8 weeks and at 12 weeks. Cost and effect results will be reported as means with SD, with mean differences between the two patient groups reported alongside 95% CIs. Depending on the amount of missing cost and quality of life data, missing data will be imputed using recommended multiple imputation methods,[50] with results from this analysis being presented as an additional sensitivity analysis.

Incremental cost-effectiveness will be calculated by dividing the difference in costs by the difference in effects. Uncertainty around the incremental cost-effectiveness ratio will be explored using non-parametric bootstrapping. A supplementary economic evaluation including non-NHS costs will be conducted in an additional sensitivity analysis. This will include costs of impaired productivity/activity and, any work and non-work days (eg, leisure or non-paid work) lost due to illness will be valued using mean average wages for those in employment. For participants who are retired or those not in employment, loss in activity levels (eg, leisure, caring or non-paid work activities) due to illness will be valued using minimum wages.

## ETHICS AND DISSEMINATION

The quality assurance evaluation of trial processes and intervention fidelity using a trial monitoring template will be carried out on 50% of the recruiting sites by the research team. Should there be no noted issues no follow-up monitoring will be conducted. There are no criteria that will flag a need to initiate a quality evaluation; however, continual scrutiny of the CRFs received from sites by the study team will alert the research team to possible issues (eg, incomplete completion of CRFs, missing data or mis-randomisation) that may subsequently require quality assurance evaluation. A standard risk assessment will be conducted and a risk-based proportionate monitoring plan will be put in place, which will include central monitoring.

Adverse event and safety oversight will comply with OCTRU SOPs. Adverse responses/reactions, adverse device effects, serious adverse responses/reactions, serious adverse device effects, suspected unexpected serious adverse responses/reactions (SUSARs) and device deficiencies will be recorded as study outcomes. Sites will be required to report all serious events that are related to a trial intervention within 24 hours and all related serious adverse events must be assessed for causality and reason for seriousness. All device deficiencies, adverse reactions and adverse device effects must be reported to the trial team within 14 days. All serious adverse event safety monitoring forms will be passed on to the Trial's Nominated Clinician who will perform an independent assessment of causality and will also perform assessment of expectedness based on what is known and documented about the intervention/device. Any SUSARs will be reported to the Research Ethics Committee within 15 days of the Trials Office being made aware of the event.

An independent DSMC (online supplementary appendix 8) will be convened with two expert clinicians and a statistician to regularly review accumulating data in order to assess patient safety and study conduct following the recommendations of the DAMOCLES study[51] with full details being provided in a charter. They will advise the Trial Steering Committee (online supplementary appendix 9) as to whether recruitment should stop early, which is only likely to occur if either intervention is shown to be unsafe for patients. As this is a low-risk study, no interim efficacy comparative analyses are planned and no safety issues are expected. The OTTER II Trial team will have access to the final trial dataset and there will be no public access to patient level data or the statistical code used within the trial.

We believe this to be the first fully powered placebo controlled splinting trial exploring the effectiveness and efficacy of splints in symptomatic thumb base OA. Study results will be disseminated to rheumatology, hand therapy, occupational therapy and physiotherapy national and international conferences and submitted for consideration in international and national academic and professional conferences and journals. Lay publications written in accessible language will be provided to charitable, community and patient facing publications as recommended by our patient and public involvement partners. Study participants will be provided regular updates of the study progress through the OTTER II study website.

**Author affiliations**
[1]School of Health Sciences. Faculty of Environment and Life Sciences, University of Southampton, Southampton, UK
[2]Nuffield Department of Orthopaedics, University of Oxford, Oxford, UK
[3]Medicine, University of Southampton, Southampton, UK
[4]Department of Life Sciences, Brunel University, Uxbridge, UK
[5]Occupational Therapy Department, Poole Hospital NHS Foundation Trust, Poole, UK
[6]Medicine, University of Nottingham, Nottingham, UK
[7]CSM, University of Oxford, OXford, UK
[8]Arthritis Research Campaign National Primary Care Centre, Keele University, Stoke on Trent, Staffordshire, UK
[9]Bristol Medical School, University of Bristol, Bristol, UK
[10]Pulvertaft Hand Centre, Royal Derby Hospital, Derby, UK
[11]Health Economics Research Centre, University of Oxford, Oxford, UK
[12]Sport, Health Sciences and Social Work Department, Oxford Brookes University, Oxford, UK

**Acknowledgements** The authors wish to acknowledge the OTTER II collaborative group: Derby Teaching Hospitals NHS Foundation Trust: Victoria Jansen, Helen McKenna, Ellen Bramall, Carole Henderson, Chloe Kirk, Diane Langford, James

Turner, Anna Selby, Linda Tozer, Navdeep Johal, Fernando Parrales, Madina Asif, Kat Hill, Stefania Wigelsworth. Doncaster and Bassetlaw Teaching Hospitals NHS Foundation Trust: Su McIlwaine, Elaine Bonser, Lynn Houghton, Ryan Roberts. Dorset County Hospital NHS Foundation Trust: Maree Dethick-Jones, Sheena Colhoun, Louise Clark, Liz Barnish, Simone Caddy, Josie Goodsell, Sarah Horton, Laura Howell, Simon Sharpe. Hampshire Hospitals NHS Foundation Trust: Kevin Spear, Christina Macleod, Libby Denman, Becky Shaylor, Kathy Whalley, Mark Pulley, Hannah Bolger. Midlands Partnership NHS Foundation Trust: Carol Graham, Nicky Edwards, Jane Rivers-Latham, Yvonne Salt, Tilly Grocott, Alison Williams, Sarah Gibson, Lisa Oakley, Susan Thompson, Susan Woodroffe, Laura Denny, Amy Thompson, Natalie Wheat. North Devon Healthcare NHS Trust: Jo Harness, Jane Hunt, Henrietta Clay, Becky Holbrook, Lucia Stancombe, Martin Howard, Nicholas McGuirk, Mark Bryce. Pennine MSK Partnership Ltd: Jill Firth, Kathy Kinsey, Helen Light, Charlotte Critchley, David Pilbury, Danielle Burke, Karen Partridge, Tracy Parry, Norah Handley, Kath Spencer. Poole Hospital NHS Foundation Trust: Sarah Bradley, Corinna Rogers, Paula Reynolds, Bridget Ellis, Sharon Page. Portsmouth Hospitals NHS Trust: Caroline Mountain, Gemma Willis, Catherine Coleman, Catherine Kirby, Paula White, Glenn Lake, Julie Williams, Marie White. Royal Devon and Exeter NHS Foundation Trust: Suzannah Blake, Abigail Owen, Emily Rogers, Claire Hughes, Cresta Browning, Jacqueline Fowler, Cristina Burke-Trees. Royal Free London NHS Foundation Trust: Juliette Bray, Nikki Burr, Meera Anadkat, Cherry Kilbride, Francesca Gowing. Royal United Hospitals Bath NHS Foundation Trust: Sandi Derham, Jessica Chipps, Suzanne Green, Helen Gordon-Johnson, Mark Sheriff, Susan Greene, Michelle Lawrence, Belinda Jones, Jack Spence, Ali Champion, Emily Anson. The Robert Jones and Agnes Hunt Orthopaedic Hospital NHS Foundation Trust: Tracy Jones, Michelle Jones, Julie Steen, Daniel Griffiths, Sally van Liefland, Jayne Edwards, Linda Griffith. Yeovil District Hospital NHS Foundation Trust: Helen Truman, Frances Campbell, Helen Jones, Jennifer Harrison, Vickie Ridley, Sue Chesterman, Joanna Allison, Tressy Pitt-Kerby, Kate Beesley. The authors also wish to acknowledge the OTTER II Trial Administrator: Carrie Fanning, University of Southampton, Southampton, UK.

**Contributors** JA: Principal investigator, conceived research project and first author of paper. PB: Trial manager, written and reviewed trial protocol and co-author. NKA: Trial team member contributed to original protocol and co-applicant and co-author. SBB: Trial team member contributed to pilot development and original protocol, co-applicant and co-author. SB: Trial team member contributed to original protocol and co-applicant and co-author. MD: Trial team member contributed to original protocol and co-applicant and co-author. SD: Trial team member contributed to original protocol, statistical lead, co-applicant and co-author. KD: Trial team member contributed to original protocol and co-applicant and co-author. RG-H: Trial team member contributed to original protocol and co-applicant and co-author. KHL: Trial team member contributed to pilot development and original protocol, co-applicant and co-author.CHG: OTTER patient partner, co-applicant, co-author, reviewed and revised paper. VJ: Trial team member contributed to revision and clinical review of original protocol and co-author. RL-F: Trial team member contributed to original protocol and co-applicant and co-author, led health economic input. CM: Trial team member, reviewed and revised qualitative component and co-author. PW: Trial team member contributed to original protocol and placebo development and co-applicant and co-author. MW: Trial team member contributed to original protocol reviewed and revised paper.

**Funding** This work was supported by Arthritis Research UK Grant Ref 21019 and the funder is listed and complies with the SHERPA JULIET guidelines http://v2.sherpa.ac.uk/id/funder/14.

**Competing interests** None declared.

**Patient consent for publication** Not required.

**Ethics approval** South Central—Oxford C Research Ethics Committee approved the study and the OTTER II Ttrial will undergo regular monitoring.

**Provenance and peer review** Not commissioned; externally peer reviewed.

**Data availability statement** There are no data in this work. All data relevant to the study are included in the article or uploaded as supplementary information.

**Open access** This is an open access article distributed in accordance with the Creative Commons Attribution 4.0 Unported (CC BY 4.0) license, which permits others to copy, redistribute, remix, transform and build upon this work for any purpose, provided the original work is properly cited, a link to the licence is given, and indication of whether changes were made. See: https://creativecommons.org/licenses/by/4.0/.

**ORCID iDs**
Jo Adams http://orcid.org/0000-0003-1765-7060

Mark Williams http://orcid.org/0000-0002-3488-847X

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
