## [Reviewer comments · BMJ Open]

ARTICLE DETAILS

TITLE (PROVISIONAL)	The Osteoarthritis Thumb Therapy Trial (OTTER II): A study protocol for a three arm multi centre randomised placebo controlled trial of the clinical effectiveness and efficacy and cost effectiveness of splints for symptomatic thumb base osteoarthritis
AUTHORS	Adams, Jo; Barratt, Paula; Arden, Nigel; Barbosa Bouças, Sofia; Bradley, Sarah; Doherty, Michael; Dutton, Susan; Dziedzic, Krysia; Gooberman-Hill, Rachael; Hislop Lennie, Kelly; Hutt Greenyer, Corinne; Jansen, Victoria; Luengo-Fernandez, Ramon; Meagher, Claire; White, Peter; Williams, Mark

VERSION 1 – REVIEW

REVIEWER	Jorge Hugo Villafañe Fondazione Don Carlo Gnocchi, Milan, Italy
REVIEW RETURNED	07-Jan-2019

GENERAL COMMENTS	INTRODUCTION: 1. Could you added comment about self-management programme? REFERENCES: 2. Would you change reference (17) with Betozzi L et al. PMID: 25559974.
---

REVIEWER	Christelle Nguyen Université Paris Descartes, France
REVIEW RETURNED	14-Jan-2019

GENERAL COMMENTS	1. GENERAL COMMENTS In this paper, authors address a major topic in the management of base-of-the-thumb osteoarthritis (OA), which is to assess the effectiveness of thumb splints when added to a self-management program in improving pain at 8 weeks for patients with symptomatic thumb OA. They also want to compare the effectiveness of verum thumb splint to placebo thumb splint and assess patients' view regarding acceptability and adherence to the trial intervention. These questions are highly relevant. However, I have concerns about: 1/ the selection of participants; 2/ the interventions tested; 3/ the reporting of outcomes and blinding. Further, the protocol is near completion of data collection. Indeed, according to the public registration (ISRCTN54744256), the study is no longer recruiting participants and will end in March 2019. Therefore, I believe my comments will unlikely be implemented in the protocol.
--

	2. MAJOR COMMENTS Selection of participants Inclusion criteria:  - Why not using the ACR and/or EULAR criteria for thumb OA? - Who will check the inclusion criteria and performed the physical examination? Exclusion criteria :  - Is it possible to include patients with chondrocalcinosis without X-ray? (4/ red flags) Interventions Self management program:  - Detailed information regarding exercises, advice and informations provided is lacking - Who will deliver the intervention: physiotherapist, occupational therapist, physician? Verum thumb splint:  - What is the Splint decision protocol? Please provide informations. Placebo thumb splint :  - According to the ISRCTN registry, participants in the placebo group will receive DM Orthotics Thumb Sleeve or DM Orthotics Thumb Sleeve Lite. Were the biomechanical and clinical effects of these orthotics sleeves previously assessed? In other words, are they placebo? Reporting of outcomes and blinding Outcomes Choice of outcome is appropriate regarding OMERACT/OARSIS guidelines. The AUSCAN hand index at 8 weeks is the primary outcome. However, there are discrepancies between the protocol in the ISRCTN registry and the paper:  - Will the AUSCAN hand index for pain be assessed at 4 weeks? Regarding secondary outcomes:  - Satisfaction with hand function over the past week at baseline, 4, 8 and 12 weeks is reported in the ISRCTN registry. Will it be assessed? - DASH questionnaire leisure function is mentioned in the paper but not in the registry. When and why was is it added? Blinding It is unclear how participants will be blinded to the intervention, especially those receiving self-management program alone. Please clarify. If blinding is impossible, there is a high risk of bias for these patients (disappointment). Why was the blinding of assessors not possible? (detection and performance bias) Other Dates of the study are not reported. 3. MINOR COMMENTS  - Patients should be 30 years old or over. Please justify why this cut off has been chosen?
--	---

	- Delay between randomization and treatment allocation is unclear (Table 2: does 0 stand for the same day? same week? other?) - Why was the GAT assessed at 4 weeks? - Exercise and splint wear diaries were completed. Does these informations used for analysis? If so, please provide information. If not, why was is recorded?
--	---

REVIEWER	Miranda Buhler University of Otago, New Zealand The first author of this paper is an international advisor for projects within my PhD programme. She has no influence over the outcome of my programme and has no financial gain from the international advisor role.
REVIEW RETURNED	21-Jan-2019

GENERAL COMMENTS	BMJOpen-2018-028342 Manuscript Review Overall the protocol outlines a study design that would allow sound interpretation of the data for the purpose of addressing the study aims. Some minor amendments to clarify the rationale and details of some aspects are recommended. TITLE The title is concise and conveys the main ideas clearly. A suggestion is, "...controlled trial of the clinical effectiveness and efficacy of splints for symptomatic thumb base osteoarthritis", rather than "for". A second suggestion is to add the cost effectiveness aspect to the title. A third suggestion is to change "thumb base osteoarthritis (OA)" to carpometacarpal osteoarthritis (CMC OA), in the title and throughout the manuscript. This would help with consistent language for the condition internationally. Whilst the protocol manuscript (pg 4) defines thumb base OA as involving "1st carpo-metacarpal and/or scaphotrapezial joint", the inclusion criteria outlined in Table 1 could include S-T joint although are arguably more specific to the CMC joint, e.g. palpation. I.e., the study findings will be mainly applicable to population with thumb CMC OA (+/- S-T OA). While reference is made to earlier publications outlining patient and public involvement (PPI), please add a short statement reporting more specifically: the results of PPI in the study protocol design, including both positive and negative outcomes; comment on the extent to which PPI influenced the study protocol overall; brief demographic information about the patients/public involved. E.g. reference #24 is a study abstract which describes a Delphi study in which 63 allied health professionals and 7 patients were asked about clinical processes and type of delivery, with some significant disagreement between AHPs and patients. What influence did the patients bring to bare on the study design? ABSTRACT The abstract is for the most part accurate, concise and well-written and is in an appropriately structured format. I would recommend mentioning the economic evaluation in the abstract as this is an important aspect of the study and would help with retrieval of articles in data base searching should systematic review of economic eval be a future review question. STRENGTHS AND LIMITATIONS An included strength is, "Trial outcome measures... agreed as meaningful by patients". The body of the manuscript (pg 5) refers to
--

a study where, “Two patient and public group meetings were carried out to inform the choice of meaningful trial outcome measures, intervention components and help to inform co-design a placebo splint design (22)”. However, while the study outlined in ref #22 aimed to involve service users in splint selection and placebo splint design to inform design of current study it did not have as its aim to identify outcome measures meaningful to patients. This study mentions fora in which participants talked about their experiences of living with thumb base OA at the beginning, however, outcome measures do not appear to have been a key topic and no conclusions were drawn about what outcome measures are considered meaningful by patients. Please either include an alternate reference for the statement (possibly ref #23) or reframe this part of the information to better match what was reported in the referenced study.

INTRODUCTION

Second paragraph (pg 4), reference #7 evidences some of the statements made in this sentence but not all – please review and add additional references as required, particularly regarding “more persistent pain, work disability, worse quality of life and function”. In third paragraph (pg 4), alludes to “traditional therapeutic splinting... is based on biomechanical principles aimed at offloading dynamic forces...”. Please can you more clearly state if this is your rationale/theory/goal of elements essential for your chosen splint intervention(s), particularly as the study design aims to elucidate efficacy as well as effectiveness. In addition, the reference given here (#11) investigated the effect of wearing a splint at night rather than use during daily tasks and makes no reference to biomechanical support offloading dynamic forces. There may be a better reference to substantiate the rationale given.

Systematic reviews referenced for evidence to date are from 2007 and 2011. A suggestion is to consider including one of several more up-to-date systematic review of studies investigating the effectiveness of this intervention.

Either in the introduction or later section, please further explain choice of comparators – i.e. the self-management package, and better describe and reference the “evidence-based exercises” (pg 7). A suggestion is that objectives (pg. 5) could include economic benefit/cost effectiveness. Please include timepoints for the first two study questions.

METHODS

Who will conduct data analysis – will they be blind to participant group allocation?

Please include a timeline for key study events, including date commence recruitment, expected date completion of recruitment. Regarding recruitment, will consecutive patients be invited? Will number and/or characteristics of those who decline to participate be recorded?

Table 1: How were clinical signs and tests selected? What is their validity and reliability in identifying people with thumb base OA? (Aside from the two reported in ref #29).

A suggestion is to give a rationale for the timeframe for exclusion criteria of intra-articular injection in previous two months.

Who will perform the eligibility screening tests and assessments?

Please give more detail about the background of the therapists delivering the interventions, e.g. physiotherapists and/or occupational therapists? Please qualify what is meant by ‘experienced in treating hand OA’. Please give more detail about the location of the intervention (NHS tertiary centres?)

Please provide reference to where data collection forms can be

	found (SPIRIT 18a). Please describe in more detail how time will be divided for each of the intervention groups, and any steps taken to maintain comparable dosage of the self-management component of the intervention, given each arm has 90 minutes direct contact time and group A will have more time available for self-management. Self-management programme – please state specific exercises, number of repetitions, frequency, and any information about grading or tailoring to individual, or provide reference to where this can be found. Please provide reference to where splint decision protocol can be found. Please state when and for how long participants are asked to wear the splint, and if any alterations are allowed. Will additional face-to-face clinical (treatment) appointments be made if needed and/or if participant requests this? Please state how the treatment period will be ended – i.e. will splints be taken back from participants at the end of the treatment period (8 weeks) or at the end of the study? Pg 9 para 4: “Training will cover clinician blinding procedures...” Does this refer to clinician procedures for maintaining participant blinding, rather than blinding of clinicians? What constitutes a requirement to conduct a quality assurance evaluation of intervention? Is there a schedule for conducting at specific intervals, or are there criteria that will flag a need? If criteria, what are they? Please describe any steps taken to prevent contamination between groups? “Good Clinical Practice training” – please reference Pg 10 para 1: “formal written consent will be...” sought, rather than given? Pg 10 para 2: Please (briefly) more clearly state how the randomisation allocation sequence will be generated, and any steps taken to conceal allocation sequence from those conducting recruitment, eligibility screening, and enrolment procedures – my apologies I am not familiar with how OCTRU facility operates. Will participants be blinded to the study hypothesis as well as to the group to which they have been allocated? Will participants know that the study is investigating the effectiveness of splints for thumb base problems? Pg 11 para 1: “good practice guidelines” – is this the same as Good Clinical Practice training? Who will perform the baseline and outcome assessments? Table 3: DASH questionnaire... Pg 14: “...Leisure section only” – does this mean the main DASH questionnaire is/not completed, and/or only the Leisure section? Will questionnaires be completed on paper or electronically? Please give the specific wording for Global assessment of change questionnaire Will all important harms or unintended effects in each group be reported as a study outcome? Pg 14 para 1: Please give reference for OTTER Pilot Study, and any other relevant sources for mean difference of 0.4 being a moderate effect size. Pg 16: Some redundancy throughout this section, e.g. paragraph 4, “...database prepared and managed by OCTRU.”, and, “...database is designed by OCTRU”. Through to page 17. Suggestion is to condense and remove repetition. However, please be more specific about data checking - who will do this? If the Statistical Analysis Plan includes other details not included in
--	---

	this protocol, please provide a reference for where these can be found. Will analysis follow 'intention-to-treat' approach? Will descriptive statistics be used to summarise group data prior to further statistical analysis and if so, which values will be presented? Pag 18: Please include more detail about the methodological approach used for the qualitative component of the study. Why will the researcher be blinded for the qualitative interviews and is this practical? It would be hard to see how the interviewer can remain blinded when the participant is describing their splint experience What is meant by "double coding" and if there is comparison between two researchers coding, how will differences be compared and/or resolved? Please provide reference for information used to base conversion of EQ5D-5L data to utilities. Pg 19 para 2: "guidelines for good economic evaluation..." – please reference Pg 20 para 2: contains timeframes for various reporting - for "Device deficiencies..." this is 2 weeks, while for SUSARs is 15 days – does this mean 15 working days, i.e. 3 weeks, or closer to 2 weeks? Please give both in either days or weeks. SPIRIT 21: "Consent will be obtained by those delegated the task by the local PI at the site." – please provide more information about who will do this – NHS therapists, other clinicians, administrative staff? Pg 21 para 1: "around thumb base OA" – please use more specific language to explain the point being made. A suggestion for dissemination is to ensure orthopaedic and plastic surgery audiences are included. MINOR GRAMMATICAL SUGGESTIONS In the first paragraph (pg 4), several different terms are used to refer to OA ("osteoarthritis", "arthritis", "the disease") and hand OA, and then thumb base OA is also introduced – more consistent language would make it clearer what is being referred to. The last sentence would benefit from re-writing for clarity. Consistency with capitalisation of 'pilot study' (p 14, 15). Pg 15: Avoid beginning sentence with RCT. Pg 20 para 2: "Device deficiencies..." - do all need capitalisation? Pg 20: REC and TSC suggest do not need abbreviating. Rather, write in full for Appendix titles. Pg 20: Last sentence, "datset" dataset Appendix 2 & 3: Recommend write TSC and DSCM in full for Appendix titles
--	---

VERSION 1 – AUTHOR RESPONSE

Reviewer: 1

Reviewer Name: Jorge Hugo Villafaña.Fondazione Don Carlo Gnocchi, Milan, Italy

INTRODUCTION:

1. Could you added comment about self-management programme?

Details about the self-management programme have now been included to the appendices as requested

REFERENCES:

2. Would you change reference (17) with Betozzi L et al. PMID: 25559974.

Delighted to update the reference and have amended as requested – thank you

Reviewer: 2

Reviewer Name: Christelle Nguyen Université Paris Descartes, France

1. GENERAL COMMENTS

In this paper, authors address a major topic in the management of base-of-the-thumb osteoarthritis (OA), which is to assess the effectiveness of thumb splints when added to a self-management program in improving pain at 8 weeks for patients with symptomatic thumb OA. They also want to compare the effectiveness of verum thumb splint to placebo thumb splint and assess patients' view regarding acceptability and adherence to the trial intervention. These questions are highly relevant. However, I have concerns about: 1/ the selection of participants; 2/ the interventions tested; 3/ the reporting of outcomes and blinding.

Further, the protocol is near completion of data collection. Indeed, according to the public registration (ISRCTN54744256), the study is no longer recruiting participants and will end in March 2019.

Therefore, I believe my comments will unlikely be implemented in the protocol.

Thank you for your feedback. You are correct that we regrettably cannot implement your comments into the protocol as these are always sent to journals for publication retrospectively. We have also had to wait until the very end of this trial to publish the protocol as we are detailing a placebo intervention. We cannot therefore publish the protocol before trial patients have completed their intervention to maintain blinding.

However, we would like to reassure the reviewers that the contribution to the design and development of the protocol has been supported by 24 independent international reviews that have already contributed to reviewing and developing the design of this protocol during the grant application process. At this stage your review and feedback is most helpful for publication and we are grateful for your time in providing your feedback.

2. MAJOR COMMENTS

Selection of participants

Inclusion criteria:

- Why not using the ACR and/or EULAR criteria for thumb OA?

The ACR/ EULAR criteria are for hand base OA not for thumb base OA, and therefore are not appropriate for this study. There are no current national /international criteria for thumb base OA in existence.

In relation to this point, reviewer number 3 also requests further detail around inclusion criteria and we have added some further detail relating to this under the inclusion criteria section. We hope this may address some of the issues around recruiting participants with symptomatic thumb base OA when there are no standardised classification criteria. No other study into symptomatic thumb base OA has used a consistent diagnostic criteria.

We hope that our baseline data can help further the discussion and debate around identifying the key characteristics of symptomatic thumb base OA and we have already arranged a Versus Arthritis summer 2019 internship project to carry out this exploration and analysis.

- Who will check the inclusion criteria and performed the physical examination?

The inclusion criteria and physical examination will be conducted by qualified clinical occupational therapists and physiotherapists (therapists) who have been trained in OTTER II Trial procedures. This detail has now been added in the script.

Exclusion criteria :

- Is it possible to include patients with chondrocalcinosis without X-ray? (4/ red flags)

The exclusion criteria has been iteratively developed over 2 years with rheumatologists, orthopaedic hand surgeons, our national expert clinicians and our research team.

We have no evidence from the design and development stage conducted for this study (where we recorded 6 month data monitoring of patients being referred for thumb base OA treatment to 5 participating therapy departments), that patients with chondrocalcinosis were being included in these referrals. Some of our patients will have already have radiological reports, all will have had medical examination and we believe that the inclusion of patients with chondrocalcinosis is highly unlikely.

Interventions

Self management program:

- Detailed information regarding exercises, advice and informations provided is lacking

We have now added further appendices of the self-management programme hand exercise and education programme as requested by all reviewers.

- Who will deliver the intervention: physiotherapist, occupational therapist, physician?

We have further clarified that the intervention will be delivered by the trained OTTER II therapy (occupational therapy and physiotherapy) clinicians.

Verum thumb splint:

- What is the Splint decision protocol? Please provide informations.

The splint decision protocol is used as a standard operating procedure to ensure that all clinicians in the trial are using a standardised approach to supplying the splints in a standardised manner across all sites. We have now included this detail as requested in the appendices.

Placebo thumb splint :

- According to the ISRCTN registry, participants in the placebo group will receive DM Orthotics Thumb Sleeve or DM Orthotics Thumb Sleeve Lite. Were the biomechanical and clinical effects of these orthotics sleeves previously assessed? In other words, are they placebo?

This is a very good question – yes these have been fully assessed in a biomechanics laboratory and we identified no known active components of the splints. We tested thumb base (CMCJ) and wrist ROM, hand function, thumb base skin surface temperature and thumb base pressure. We detected no trends nor clinically or statistically significant data that indicate that the placebo splints have any measurable effect on the parameters tested.

We could not publish anything about the placebo until we had completed patient recruitment and follow up. However, the following abstract has been accepted at BSR 2019 (Loyley, M., Davies, L., Worsley P., Adams J Comparison of the functional impact of verum and placebo thumb base orthoses: a proof of concept study) and we now add this as an in press publication to our reference list. We currently also have a further abstract in review for EULAR 2019 (Davies L., Loyley, M., Worsley, P., Adams, J., Comparison of the impact on skin surface temperature and pressure exerted over the carpometacarpal joint of verum and placebo thumb base orthoses) should this be accepted prior to this protocol being published we will also be able to add this to our reference list.

Reporting of outcomes and blinding

Outcomes

Choice of outcome is appropriate regarding OMERACT/OARSIS guidelines. The AUSCAN hand index at 8 weeks is the primary outcome. However, there are discrepancies between the protocol in the ISRCTN registry and the paper:

- Will the AUSCAN hand index for pain be assessed at 4 weeks?

Thank you. This discrepancy has now been updated in the ISRCTN registry. The protocol paper details the correct assessment timepoints for AUSCAN.

Regarding secondary outcomes:

- Satisfaction with hand function over the past week at baseline, 4, 8 and 12 weeks is reported in the ISRCTN registry. Will it be assessed?

The information on the ISRCTN registry is as the following screenshot details. We have no assessment at 4/52 and this has not been included :

As we have detailed in the paper formal assessment using standardised satisfaction with hand function questions from the Michigan Hand Outcomes Questionnaire is included at Baseline, 8 and 12 weeks. This is consistent with what is stated in our protocol paper and what is reported in our trial registry details. If there is any other discrepancy please do let us know and we can address this, however, we believe this issue to be consistent throughout.

- DASH questionnaire leisure function is mentioned in the paper but not in the registry. When and why was it added?

We believe the details are already consistent with our ISRCTN registry data and we include below a screen shot of what is included in the trial registry data. This is consistent with the details we provide in our protocol paper:

If there is any other discrepancy please do let us know and we can address this, however, we believe this issue to be consistent throughout.

Blinding: It is unclear how participants will be blinded to the intervention, especially those receiving self-management program alone. Please clarify. If blinding is impossible, there is a high risk of bias for these patients (disappointment).

We agree blinding is a vital component of our trial. Blinding has been successfully achieved in our pilot study and the same strategies are used in the OTTER II Trial. We address these issues by:

- 1 Training of the clinicians to knowingly delivering placebo treatment without disclosing or giving hints to treatment arm to participating patients
- 2 Careful wording of the participant information sheet (approved by our ethical committee) that details that an element of the treatment may be placebo – but does not detail what component may be.

3 Participants are not informed of what the self-management treatment package comprises in each arm. Splinting is a part of this overall self-management package in two intervention arms. They are therefore blind to the contents of each arm.

4 Patients are not treated together so cannot see what others in the trial have been given.

5 Excluding patients who are not splint naïve; i.e. who have previously had splints to treat their thumb arthritis

6 We have designed the trial to minimise any patient disappointment in being allocated to the self-management only (minus a splint) arm as the time provided by the treating therapist is the same as that provided to the self-management plus splint treatment arm.

7 We have closely monitored the number of patients who have declined to take part in the OTTER II study for a variety of reasons. We do not have evidence that being disappointed to being allocated to self-management alone (or not receiving a splint) has been a significant issue.

We agree that these are important issues that can be fully discussed in the results paper when ready for publication. We hope the details above address your concerns on this issue. Some of the detail relating to the success of blinding we cannot include in a prospective protocol paper as this belongs in the full trial reporting paper. However, we have now included further detail as requested relating to patient blinding in the protocol

Why was the blinding of assessors not possible? (detection and performance bias)

Assessors are fully blinded to treatment allocation. It is only the treating OTTER II clinicians (who are not the assessors) who deliver the interventions and knowingly deliver the placebo options that cannot be blinded. We have ensured that this is now further clarified in the paper.

Other

Dates of the study are not reported.

Thank you these are now included from first patient first visit to last patient last follow up.

3. MINOR COMMENTS

- Patients should be 30 years old or over. Please justify why this cut off has been chosen?

Our inclusion criteria were decided upon iteratively with our rheumatologists, clinicians and hand surgeons in relation to the literature about prevalence, incidence rates and treatment pathways to therapy departments in the UK. A sentence to this effect has been added at the start of the inclusion criteria section.

- Delay between randomization and treatment allocation is unclear (Table 2: does 0 stand for the same day? same week? other?)

We have now further clarified the table to make this clearer. 0 stands for Baseline i.e. Day 0 and is the same day.

- Why was the GAT assessed at 4 weeks?

The GAT was used as a short term measure of “objective” hand function at 4 weeks as opposed to PROMs. It was reported upon by our patient partners and clinical collaborators as a useful opportunity to help maintain further meaningful clinical contact for patients travelling in for additional clinic appointments so was included. The data will be included for analysis.

- Exercise and splint wear diaries were completed. Does these informations used for analysis? If so, please provide information. If not, why was is recorded?

Yes this is an excellent question and adherence is an area that has generated a lot of debate and discussion in our team. We will be using this data for secondary per protocol analysis and we have now included a statement to this effect. All data collected will contribute towards analysis.

Reviewer: 3

Reviewer Name: Miranda Buhler University of Otago, New Zealand

Please state any competing interests or state 'None declared': The first author of this paper is an international advisor for projects within my PhD programme. She has no influence over the outcome of my programme and has no financial gain from the international advisor role.

Overall the protocol outlines a study design that would allow sound interpretation of the data for the purpose of addressing the study aims. Some minor amendments to clarify the rationale and details of some aspects are recommended.

TITLE

The title is concise and conveys the main ideas clearly. A suggestion is, "...controlled trial of the clinical effectiveness and efficacy of splints for symptomatic thumb base osteoarthritis", rather than "for".

Thank you we have made this suggested change

A second suggestion is to add the cost effectiveness aspect to the title.

Thank you we have made this suggested change

A third suggestion is to change "thumb base osteoarthritis (OA)" to carpometacarpal osteoarthritis (CMC OA), in the title and throughout the manuscript. This would help with consistent language for the condition internationally. Whilst the protocol manuscript (pg 4) defines thumb base OA as involving "1st carpo-metacarpal and/or scaphotrapezial joint", the inclusion criteria outlined in Table 1 could include S-T joint although are arguably more specific to the CMC joint, e.g. palpation. I.e., the study findings will be mainly applicable to population with thumb CMC OA (+/- S-T OA).

Thank you for the suggestion – we agree this is perfectly relevant to consider and has been an ongoing discussion with the team and PPI partners. We have revisited this again following your comments. We think that the term thumb base OA includes both CMC OA and S-T OA (which reflects the patient population that are routinely treated in our recruiting centres) and also describes the symptomatic nature of our target population. We also believe this to be more simple that stating CMC OA +/-S-T OA each time when describing our population. Following discussion with colleagues we also think that in searching for the paper that both TBOA and CMC OA would be used. We appreciate that different clinical groups use different terminology and within rheumatology (where our funding came from) TBOA was deemed to be appropriate terminology by our funders and international review group who sanctioned the funding of this research.

While reference is made to earlier publications outlining patient and public involvement (PPI), please add a short statement reporting more specifically:

- the results of PPI in the study protocol design,
- including both positive and negative outcomes;
- comment on the extent to which PPI influenced the study protocol overall;
- brief demographic information about the patients/public involved.
- E.g. reference #24 is a study abstract which describes a Delphi study in which 63 allied health professionals and 7 patients were asked about clinical processes and type of delivery, with some significant disagreement between AHPs and patients
- What influence did the patients bring to bare on the study design?

We have added some further detail about the involvement of patients in designing the study as suggested on pp5-6.

ABSTRACT

The abstract is for the most part accurate, concise and well-written and is in an appropriately structured format. I would recommend mentioning the economic evaluation in the abstract as this is an important aspect of the study and would help with retrieval of articles in data base searching should systematic review of economic eval be a future review question.

Cost effectiveness is now included in the abstract

STRENGTHS AND LIMITATIONS

An included strength is, “Trial outcome measures... agreed as meaningful by patients”. The body of the manuscript (pg 5) refers to a study where, “Two patient and public group meetings were carried out to inform the choice of meaningful trial outcome measures, intervention components and help to inform co-design a placebo splint design (22)”. However, while the study outlined in ref #22 aimed to involve service users in splint selection and placebo splint design to inform design of current study it did not have as its aim to identify outcome measures meaningful to patients. This study mentions fora in which participants talked about their experiences of living with thumb base OA at the beginning, however, outcome measures do not appear to have been a key topic and no conclusions were drawn about what outcome measures are considered meaningful by patients.

Please either include an alternate reference for the statement (possibly ref #23) or reframe this part of the information to better match what was reported in the referenced study.

We have amended this PPI section as requested and better aligned the reference to meaningful patient outcomes to Ref 23.

INTRODUCTION

Second paragraph (pg 4), reference #7 evidences some of the statements made in this sentence but not all – please review and add additional references as required, particularly regarding “more persistent pain, work disability, worse quality of life and function”.

The word “persistent” has been removed and 3 additional references are now added to this section to fully support the statement.

In third paragraph (pg 4), alludes to “traditional therapeutic splinting... is based on biomechanical principles aimed at offloading dynamic forces...”. Please can you more clearly state if this is your rational/theory/goal of elements essential for your chosen splint intervention(s), particularly as the study design aims to elucidate efficacy as well as effectiveness. In addition, the reference given here (#11) investigated the effect of wearing a splint at night rather than use during daily tasks and makes

no reference to biomechanical support offloading dynamic forces. There may be a better reference to substantiate the rationale given.

Our pragmatic trial has been informed and developed with patient partners and clinicians and is designed to run across 17 NHS hospital sites in the UK. The rationale behind traditional therapeutic splinting is indeed our starting point for examining what “active ingredients” may be present in splinting. However, we cannot state that this theory was “essential” in our chosen splints. Our basis for the design and development of our trial was as a collaborative iterative process with PPI partners and clinicians. No verum splint designs or options were therefore excluded from them to consider to include in our trial. Placebo splint designs were developed with patient, clinician and industry input with no known active component. Upon testing (to be reported elsewhere and not part of this protocol paper) we are able to present data that confirms this (Please also see response to reviewer no. 2 above).

We have re-worded the section on splinting background, provided further references and added a short statement to state that examining the efficacy of traditional biomechanical approach to splinting formed the rationale for this study as suggested.

Systematic reviews referenced for evidence to date are from 2007 and 2011. A suggestion is to consider including one of several more up-to-date systematic review of studies investigating the effectiveness of this intervention.

Thank you this is in line with another reviewer’s comments and we have amended as reviewer no. 1 has requested.

Either in the introduction or later section, please further explain choice of comparators – i.e. the selfmanagement package, and better describe and reference the “evidence-based exercises” (pg 7).

Details for the interventions are now included as appendices and we have added the reference to the evidence based systematic review on which the OTTER II standardised exercises were based.

A suggestion is that objectives (pg. 5) could include economic benefit/cost effectiveness.

Thank you for this comment we agree this could be included but the original objectives agreed by the funding organisation followed international review and revision and are as we have stated in the paper. We have amended the abstract to include cost effectiveness which we believe is helpful. We would prefer to keep the objectives here as agreed with our funders.

Please include timepoints for the first two study questions.

It is unusual to provide timepoints for the study questions per se, as these will be answered following analysis of the data, in line with other trials. What we believe to be more informative is the addition of timelines for the study (as requested by reviewer no. 2) and we have now added these. We hope this also addresses what you are seeking.

METHODS

Who will conduct data analysis – will they be blind to participant group allocation?

The OCTRU trial statistician will conduct the data analysis and will not be blind to treatment allocation. This is usual practice for the Oxford Clinical Trials Unit for rehabilitation RCTs. There are rigorous checks and balances in place to ensure that the statistician is unable to bias outcome. We have added a short statement to this effect at the start of the statistical analysis section.

Please include a timeline for key study events, including date commence recruitment, expected date completion of recruitment.

Timepoints for the study have now been included from first patient first visit to last patient last follow up visit.

Regarding recruitment, will consecutive patients be invited?

Yes, this is correct and is now detailed in the text.

Will number and/or characteristics of those who decline to participate be recorded?

Yes, this is usual practice and a sentence is now added to clarify this.

Table 1: How were clinical signs and tests selected?

There is no agreed classification for thumb base OA currently available. Hand OA has diagnostic criteria already established. In this scenario we had to agree the study's own clinical tests within the trial team including dialogue with expert clinicians.

Our clinical signs and tests were selected following:

- 1 A review of the literature of available tests that have been published
- 2 Consideration of the strength of evidence (including expert opinion) of sensitivity and specificity
- 3 An examination of the practicality and feasibility of the testing procedure for a) the collaborating clinicians who will be screening and recruiting OTTER patients and b) the 17 different recruiting hospital pathways for thumb base OA pts as the procedure had to be practical and pragmatic for each.

Our team made a considered decision not to include the potentially painful grind test (Merritt 2010) at the start of a trial that required collaboration and cooperation from patients. This was universally supported by our collaborating clinicians and research team. Our collaborating therapists and rheumatologists, did not want to use this test.

Please also refer to our comments to reviewer no. 2 regarding a related issue. We have added additional detail to the paper in line with this.

What is their validity and reliability in identifying people with thumb base OA? (Aside from the two reported in ref #29).

Our trial is concerned with identifying people with symptomatic thumb base OA. We are not funded to identify people with radiographic TBOA and our focus is on symptomatic presentation as this is the population treated by our therapists. All of the signs included were agreed as having the potential to identify people with symptomatic TBOA i.e. they had agreement with the study team and expert collaborating clinicians as having the potential to be specific and sensitive. These tests had also been evident in previous literature as having been used in identifying patients with symptomatic TBOA. We believe that in the absence of diagnostic criteria this was the most robust approach available to us and defend our approach.

Currently there is limited evidence for validity and reliability of these clinical measures, (other than those you identify.). However, our OTTER II trial training provided guidance and practical training to address all aspects of patient assessment and treatment including practical workshops on

administering these tests, as detailed in the text. We believe that this addresses some of the inter and intra-rater reliability issues across recruiting sites. Additionally all our quality assurance visits carried out on 50% of the recruiting sites are a second check to ascertain that procedures are being conducted in a similar (i.e. reliable manner). These were agreed and authorised by OCTRU as adhering to usual trial expected standards carried out in the NHS in England.

Detail relating to training and QA visits is included in the manuscript. We believe that these measures are beyond those used by other trials examining symptomatic TBOA conservative interventions and we anticipate that the large data set (n=345) we will collect will serve to further the debate and discussion around diagnosing symptomatic TBOA. A 2019 summer internship project with Versus Arthritis has already been agreed to do this – but this detail is beyond the scope of a protocol publication.

A suggestion is to give a rationale for the timeframe for exclusion criteria of intra-articular injection in previous two months.

The rationale was based on the fact that there has been no clear evidence nor consensus for the length of time that intra-articular injections to the CMC are effective or efficacious in reducing pain levels (Trellau et al 2015; Meehagh et al 2004; Heyworth et al 2008, Kroon et al 2016). Pain relief from intra-articular injections can happen within first 4 weeks but could last up to several weeks (Gossec et al 2004; Heyworth, et al. 2008). Clinically if after 2 months patients are still presenting with hand pain and hand dysfunction following an intra-articular injection then they would still be eligible to take part.

With no clear guidelines on the longevity of intra articular steroids for thumb base OA we referred to

- i) expert clinician advice relating to local (non-referenced) protocols for IA injections not to be repeated within 6/52
- ii) a review of patient characteristics from our screening logs from our pilot study and our recruitment rate to recruit sufficient patients within our funded timelines to be fully powered.

For those patients who may derive longer lasting effect from an IA injection this will be balanced across treatment groups.

Who will perform the eligibility screening tests and assessments?

Trained collaborating OTTER II clinicians will perform the eligibility screening tests and assessments and this is now clarified

Please give more detail about the background of the therapists delivering the interventions, e.g. physiotherapists and/or occupational therapists? Please qualify what is meant by 'experienced in treating hand OA'.

We have provided further detail that the intervention is delivered by a qualified OTTER II trained occupational therapist or physiotherapist who have worked independently in a clinical role treating patients with hand OA and who works within a therapy department that accepts referrals for thumb base OA patients.

Please give more detail about the location of the intervention (NHS tertiary centres?)

We have added detail about the site being an NHS secondary care hospital or clinic

Please provide reference to where data collection forms can be found (SPIRIT 18a).

These can be obtained from the first author and these details are now added.

Please describe in more detail how time will be divided for each of the intervention groups, and any steps taken to maintain comparable dosage of the self-management component of the intervention, given each arm has 90 minutes direct contact time and group A will have more time available for self-management.

The self-management intervention will be longer in the self-management only group as the time with a therapist is comparable across treatment arms. Within each arm clinicians will deliver the standardised interventions, now detailed in the appendices, within 90mins. Should we not have designed a trial with this approach the face to face time with a therapist would be different between each treatment arms which the team universally decided was not desirable.

It is not possible to state how much time will be spent on each component as the self-management interventions are applied to each patient's needs using OTTER trained clinician's clinical judgement alongside the standardised booklet information. To allocate specific times to each part of the self-management programme would be arbitrary, not tailored to patient centred care and not feasible to deliver within a pragmatic rehabilitation RCT.

We have now added a statement relating to this in the text.

Self-management programme – please state specific exercises, number of repetitions, frequency, and any information about grading or tailoring to individual, or provide reference to where this can be found.

Appendices for the self-management programme are now included

Please provide reference to where splint decision protocol can be found.

Appendix for the splint decision protocol is now included

Please state when and for how long participants are asked to wear the splint, and if any alterations are allowed.

Appendix for the splint wear instructions is now included

Will additional face-to-face clinical (treatment) appointments be made if needed and/or if participant requests this?

No, the treatment provided in the trial is what is currently specified in the paper.

Please state how the treatment period will be ended – i.e. will splints be taken back from participants at the end of the treatment period (8 weeks) or at the end of the study?

We have added further detail to our end of study procedures in the text in relation to splints as requested.

Pg 9 para 4: "Training will cover clinician blinding procedures..." Does this refer to clinician procedures for maintaining participant blinding, rather than blinding of clinicians?

Thank you now amended as suggested.

What constitutes a requirement to conduct a quality assurance evaluation of intervention?
Is there a schedule for conducting at specific intervals, or are there criteria that will flag a need? If criteria, what are they?

The quality assurance evaluation of trial processes and intervention fidelity using a trial monitoring template will be carried out on 50% of the recruiting sites by the research team. Should there be no noted issues no follow-up monitoring will be conducted. There are no criteria that will flag a need to initiate an evaluation, however, continual scrutiny of the CRFs received from sites will alert the research team to possible issues (e.g. incomplete completion of CRFs, missing data or mis-randomisation) that may subsequently require quality assurance evaluation. Detail regarding this is now added to the text.

Please describe any steps taken to prevent contamination between groups?

The main area for possible contamination is participants accessing and wearing verum splints if allocated to either Arm A or Arm B. The placebo splints cannot be openly purchased or obtained as are only manufactured for the OTTER II Trial. We cannot prevent people accessing and using thumb splints on the open market as these are commercially available. We do have a mechanism for recording this in that we will ask if splints have been bought and used during the trial. We may also capture this information through the qualitative interviews. All disclosed incidences when patients buy and wear verum splints will be recorded and will contribute to analysis. We have added some further detail relating to this in the text.

“Good Clinical Practice training” – please reference

A reference is now provided for GCP training

Pg 10 para 1: “formal written consent will be...” sought, rather than given?

We have amended this to “sought and given” – as we cannot proceed unless consent is given so this needs to be included.

Pg 10 para 2: Please (briefly) more clearly state how the randomisation allocation sequence will be generated, and any steps taken to conceal allocation sequence from those conducting recruitment, eligibility screening, and enrolment procedures – my apologies I am not familiar with how OCTRU facility operates.

Details are now included regarding randomisation sequence to clarify these issues as requested

Will participants be blinded to the study hypothesis as well as to the group to which they have been allocated?

Yes we have carefully worded the PIS and have included further details re maintaining participant blinding in the text in line with reviewer no 2.

Will participants know that the study is investigating the effectiveness of splints for thumb base problems?

No, we have detailed this in our carefully worded PIS that has received full NHS ethical scrutiny and approval.

We have added further detail regarding participant blinding in the text in line with reviewer's no 2 request.

Pg 11 para 1: "good practice guidelines" – is this the same as Good Clinical Practice training? Who will perform the baseline and outcome assessments?

Thank you for this. We have now deleted the statement referring to good practice guidelines so as not to contribute to any confusion with GCP.

Table 3: DASH questionnaire... Pg 14: "...Leisure section only" – does this mean the main DASH questionnaire is/not completed, and/or only the Leisure section?

We have asked participants to complete the "Leisure section only" as we already state in the text. We hope that this is clear for readers.

Will questionnaires be completed on paper or electronically?

We have already detailed that questionnaires are "posted" to participants and have added "paper" to the text to ensure that this is clear.

Please give the specific wording for Global assessment of change questionnaire

This question and it's VAS response is now added under Table 3 as requested

Will all important harms or unintended effects in each group be reported as a study outcome?

Yes this is standard and accepted practice for the OCTRU and is included under the section detailed as adverse events on page 22. Important harms and unintended events that are deemed to be serious are classified under this system. We have added "as a study outcome" to the section to clarify this.

Pg 14 para 1: Please give reference for OTTER Pilot Study, and any other relevant sources for mean difference of 0.4 being a moderate effect size.

The pilot study is originally referenced as ref No 31 . We now re-state this in the statistical section for further clarity.

A reference for effect size is now included.

Pg 16: Some redundancy throughout this section, e.g. paragraph 4, "...database prepared and managed by OCTRU.", and, "...database is designed by OCTRU". Through to page 17. Suggestion is to condense and remove repetition. However, please be more specific about data checking - who will do this?

This section has been re-written to reduce any repetition.

If the Statistical Analysis Plan includes other details not included in this protocol, please provide a reference for where these can be found.

The SAP has not yet been published and currently does not include further detail than that included here.

Will analysis follow 'intention-to-treat' approach?

Yes the main analysis will be ITT and this is now clarified in the text.

Will descriptive statistics be used to summarise group data prior to further statistical analysis and if so, which values will be presented?

Yes appropriate descriptive analysis will be used. We will use the appropriate methods according to distribution and have added a sentence to this effect. We feel it would be unusual in a protocol paper to list the various combinations of descriptive statistics that could possibly be used and hope that the sentence above is sufficient.

Pag 18: Please include more detail about the methodological approach used for the qualitative component of the study.

We have included further detail as requested

Why will the researcher be blinded for the qualitative interviews and is this practical? It would be hard to see how the interviewer can remain blinded when the participant is describing their splint experience

The researcher will be blinded for the qualitative interviews in order to prevent bias related to treatment allocation during data collection. In order to make this practical, the researcher carrying out data collection will be independent from the RCT and will be unfamiliar with the splints and differences between them. As the splints are aesthetically identical no information regarding the appearance of the splint or other factors could unblind the researcher. We have now added this detail to the text

What is meant by "double coding" and if there is comparison between two researchers coding, how will differences be compared and/or resolved?

Double coding means more than one researcher independently assigning pre-specified codes to the qualitative data. NVivo software (QSR International) will be used to facilitate the double coding process. This allows for coding comparisons to be run between different coders. The results are returned according to the Kappa coefficient score and the percentage of agreement between coders. After double coding transcripts, we plan to use this coding comparison for discussion and reflection on the data. The numeric measure will be used as a method of comparison to gauge if there is agreement and understanding of the definitions of the codes between team members. Any discrepancies identified will then be discussed until an agreement is reached and any differences resolved.

This detail is now added to the paper.

Please provide reference for information used to base conversion of EQ5D-5L data to utilities.

This is now included as requested

Pg 19 para 2: "guidelines for good economic evaluation..." – please reference

This is now included as requested

Pg 20 para 2: contains timeframes for various reporting - for “Device deficiencies...” this is 2 weeks, while for SUSARs is 15 days – does this mean 15 working days, i.e. 3 weeks, or closer to 2 weeks? Please give both in either days or weeks.

Thank you – this is now amended and consistently reported throughout

SPIRIT 21: “Consent will be obtained by those delegated the task by the local PI at the site.” – please provide more information about who will do this – NHS therapists, other clinicians, administrative staff?

This detail has now been added as requested

Pg 21 para 1: “around thumb base OA” – please use more specific language to explain the point being made.

Thank you this is now re-written

A suggestion for dissemination is to ensure orthopaedic and plastic surgery audiences are included.

Thank you we note your suggestion but would like to keep the text as it currently is. We believe this to be an inclusive statement without naming lists of specific clinical groups. This is what we intend.

MINOR GRAMMATICAL SUGGESTIONS

In the first paragraph (pg 4), several different terms are used to refer to OA (“osteoarthritis”, “arthritis”, “the disease”) and hand OA, and then thumb base OA is also introduced – more consistent language would make it clearer what is being referred to. The last sentence would benefit from re-writing for clarity.

We have changed “disease” to osteoarthritis. However, we would like to keep the text that we originally propose as hand OA is different from thumb base OA and we are introducing different concepts. We are unable to see a way to make these the same.

Consistency with capitalisation of ‘pilot study’ (p 14, 15).

Amended thank you

Pg 15: Avoid beginning sentence with RCT.

Amended thank you

Pg 20 para 2: “Device deficiencies...” - do all need capitalisation?

These do need capitalisation as they are formal heading titles so we would like to remain as originally stated.

Pg 20: REC and TSC suggest do not need abbreviating. Rather, write in full for Appendix titles.

Amended thank you

Pg 20: Last sentence, "dataset" dataset

Amended thank you

Appendix 2 & 3: Recomme

Our apologies but your comments appear to be truncated as this seems to have been cut off when sent across – this is the last sentence we received.

VERSION 2 – REVIEW

REVIEWER	Christelle Nguyen Université Paris Descartes, Faculté de Médecine, Sorbonne Paris Cité
REVIEW RETURNED	13-Apr-2019

GENERAL COMMENTS	I want to thank the authors for their detailed and satisfying answers and appropriate changes and statements in both manuscript and appendices. However, regarding the dates reported in the final manuscript (from 28th February 2017 to 14th March 2019) it seems that data collection is now completed, which might be inconsistent with BMJ open's instructions regarding protocol papers, only for planned or ongoing studies. I already had raised this concern, to what the authors answered that the protocol publication has been delayed until the end of the trial because they were detailing a placebo intervention. I think this answer might alter the confidence we can place into the placebo intervention. Indeed, if the placebo intervention is a real placebo and if blinding and allocation concealment are done properly, knowing the nature of the placebo should not influence the results. However, in non-pharmacological intervention, placebo and blinding are always challenging, and I want to thank the authors for addressing this major topic in the management of hand osteoarthritis.
---

REVIEWER	Miranda Buhler University of Otago
	The first author is an international advisor for aspects of my PhD programme. However, she has no influence over the outcome of my programme and has no financial gain from the international advisor role.
REVIEW RETURNED	26-Mar-2019

GENERAL COMMENTS	Thank you, most comments and issues raised have been well addressed. Some minor comments are as follows: The protocol describes the intervention period as 8 weeks (Pg 6), whereas Appendices 3, 4 and 7 describe, "splint to wear for 12 weeks", and Appendices 2 and 6 outline self-management and splint-wearing diaries lasting for 12 weeks, suggesting the intervention period was 12 rather than 8 weeks?
---

	The sentence regarding written consent (Pg 11) has been amended, thank you, however, perhaps rather than “given” and “taken, it would better reflect the informed consent process to use ‘obtained’ in both sentences. Trial Design and Setting section describes trial as, “single (participant) blind” (Pg 6), whereas section “Treatment and follow up” outlines, “an assessor blind to treatment allocation will administer the objective Grip Ability Test” (Pg 13). Given all other outcome assessments are patient-reported, and the patient is blinded, and they are their own assessor, then it could be argued the study is double-blind. Or at least partially assessor-blind (for objective measure). Per your author response, “Assessors are fully blinded to treatment allocation”. Pg 11, 3rd line, “... procedures for maintaining clinician blinding...” – this sentence perhaps not quite amended as intended? Reference #20, for insufficient robust research to support effectiveness of splints is from 2007 – could be updated with Betozzi L et al. (2015) PMID: 25559974, or Aebischer B et al. (2015) PMID: 27110291, or Buhler M et al. (Epub 2018) PMID: 30317000. Some minor grammatical errors, e.g. last line Pg 5; 4th-to-last line Pg 4. The details of who will conduct the intervention has been added twice (Pg 6-7 and Pg 8) – recommend delete from Pg 6-7 and leave in section ‘Interventions’.
--	--

VERSION 2 – AUTHOR RESPONSE

Reviewer: 3

Thank you, most comments and issues raised have been well addressed. Some minor comments are as follows:

The protocol describes the intervention period as 8 weeks (Pg 6), whereas Appendices 3, 4 and 7 describe, “splint to wear for 12 weeks”, and Appendices 2 and 6 outline self-management and splint-wearing diaries lasting for 12 weeks, suggesting the intervention period was 12 rather than 8 weeks?

Thank you – this is helpful. The difference relates to the 8/52 intervention period and 4/52 follow up period. We are able to understand how this could appear confusing and have amended page 6 to clarify that whilst facilitated self-management will continue for 8/52 participants will be encouraged to continue their intervention until 12 weeks from baseline.

The sentence regarding written consent (Pg 11) has been amended, thank you, however, perhaps rather than “given” and “taken, it would better reflect the informed consent process to use ‘obtained’ in both sentences.

Happy to amend as suggested and obtained is now inserted.

Trial Design and Setting section describes trial as, “single (participant) blind” (Pg 6), whereas section “Treatment and follow up” outlines, “an assessor blind to treatment allocation will administer the objective Grip Ability Test” (Pg 13). Given all other outcome assessments are patient-reported, and the patient is blinded, and they are their own assessor, then it could be argued the study is double-

blind. Or at least partially assessor-blind (for objective measure). Per your author response, "Assessors are fully blinded to treatment allocation".

We are very reluctant to class this as a double blind study as the clinicians providing the intervention cannot be blinded. We have been incredibly careful to train and support clinicians in being objective in their delivery but we cannot provide evidence that their knowledge in knowingly delivering a placebo splint did not influence self-reported outcome in some way. You are correct in acknowledging that the assessor is blind.

A double blind study is where both patient and clinician are blind – this is not the case with OTTER and we have considered very carefully how we define our study and as such prefer to keep our terminology as stated. This is also what has been agreed with our funders.

Pg 11, 3rd line, "... procedures for maintaining clinician blinding..." – this sentence perhaps not quite amended as intended?

Thank you ! Amended to reflect what was intended

Reference #20, for insufficient robust research to support effectiveness of splints is from 2007 – could be updated with Betozzi L et al. (2015) PMID: 25559974, or Aebischer B et al. (2015) PMID: 27110291, or Buhler M et al. (Epub 2018) PMID: 30317000.

We had already updated the reference list to include Bertozzi et al 2015 as requested by reviewer 1 so this was already inserted into our revised manuscript. When we first wrote this paper your review had not been published so we could not use your paper. However, we have now inserted this to our reference list too.

Some minor grammatical errors, e.g. last line Pg 5; - now amended; 4th-to-last line Pg 4.now amended

The details of who will conduct the intervention has been added twice (Pg 6-7 and Pg 8) – recommend delete from Pg 6-7 and leave in section 'Interventions'.

We have added the additional details following a review request from another reviewer. As your request to delete this detail will override a previous request to add this detail we are happy to leave this editing decision to the editor. We have not amended the text here.

Reviewer: 2

Please leave your comments for the authors below

I want to thank the authors for their detailed and satisfying answers and appropriate changes and statements in both manuscript and appendices.

Thank you for your comment, this is much appreciated.

However, regarding the dates reported in the final manuscript (from 28th February 2017 to 14th March 2019) it seems that data collection is now completed, which might be inconsistent with BMJ open's instructions regarding protocol papers, only for planned or ongoing studies.

When we submitted the protocol to BMJ Open we were still treating trial patients and treatment was still ongoing within the trial. This is within the remit of the authors guidelines for protocols.

I already had raised this concern, to what the authors answered that the protocol publication has been delayed until the end of the trial because they were detailing a placebo intervention. I think this answer might alter the confidence we can place into the placebo intervention. Indeed, if the placebo

intervention is a real placebo and if blinding and allocation concealment are done properly, knowing the nature of the placebo should not influence the results.

Thank you for your comment and we can understand your query. However, if patients become unblinded by sourcing the information via an academic paper then we do believe that this could impact on patient reporting of any impact a placebo may have. As PI I have been very keen to ensure that nothing is presented/published that may undermine our determined effects to maintain blinding to placebo interventions. I would be unable to confidently defend this if there is a paper out that details which design is a placebo. We have carefully timed our publications.

However, we have referenced our placebo publication in the reference list now (BSR finished only 6 days ago) and hope that this provides some confidence for the readers that these do have evidence that to our current knowledge, these placebo splints have no measurable impact on hand function.

However, in non-pharmacological intervention, placebo and blinding are always challenging, and I want to thank the authors for addressing this major topic in the management of hand osteoarthritis.

Thank you OTTER II has been a bold trial that has successfully recruited to time and target due to the terrific clinical and academic teams we have engaged. I am sure it will encourage ongoing debate !